# ParaThinker: Native Parallel Thinking as a New Paradigm to Scale LLM Test-time Compute

## Abstract

Recent advances in Large Language Models (LLMs) have been driven by test-time compute scaling - a strategy that improves reasoning by generating longer, sequential thought processes. While effective, this approach encounters a significant bottleneck as computation increases, where further computation offers only marginal performance gains. We argue that this ceiling is not an inherent limit of the model's capability but a flaw in the scaling strategy itself, a phenomenon we term *"Tunnel Vision"*, where a model's imperfect initial steps lock it into a suboptimal reasoning path. To overcome this, we introduce a new scaling paradigm: *native thought parallelism*. We present *ParaThinker*, an end-to-end framework that trains an LLM to generate multiple, diverse reasoning paths in parallel and synthesize them into a superior final answer. By exploring different lines of thought simultaneously, *ParaThinker* effectively sidesteps the *Tunnel Vision* issue and unlocks the model's latent reasoning potential. Our approach demonstrates that scaling compute in parallel (width) is a more effective and efficient way to superior reasoning than simply scaling sequentially (depth). On challenging reasoning benchmarks, ParaThinker achieves substantial accuracy improvements (e.g. 6.5%-20.7% on AIME-24 with 1.5B model) over sequential-reasoning LLMs under the same budget of decoding tokens, without inducing additional computational cost. The reasoning latency can even be reduced by 38.7%-66.8% via batch decoding in on-device single-request settings.

## 1 Introduction

The remarkable progress of Large Language Models (LLMs) has been largely driven by the principle of scaling. This evolution began with pretraining compute scaling and has recently shifted to post-training or test-time compute scaling. Notable examples of test-time scaling, such as OpenAI o1 (OpenAI, 2024) and DeepSeek-R1 (DeepSeek-AI, 2025), have demonstrated that training the models to "think longer" (*i.e.* decode more tokens before generating the final answer) can unlock superior reasoning abilities for complex problems (Yang et al., 2025a; Team et al., 2025; Snell et al., 2024; Wu et al., 2025).

However, extending test-time compute does not lead to constant performance improvement in today's reasoning LLMs, where accuracy improvements diminish and eventually stagnate after a certain number of decoding steps. This has fueled discussions around "LLM overthinking" (Ghosal et al., 2025; Chen et al., 2025b; Fan et al., 2025; Cuadron et al., 2025; Li et al., 2025), where models expend excessive computation on problems, with the additional reasoning steps yielding minimal or no benefit to the final answer.

In this paper, we investigate the problem of test-time scaling bottleneck by raising a fundamental question: *Does the test-time scaling bottleneck stem from the inherent limitations of the model's capability, or from the imperfect test-time compute strategy?* The answer to this question is important for understanding the bottleneck of test-time scaling. Our findings reveal that, given a fixed decoding token budget, the conventional self-refinement reasoning paradigm (adopted in state-of-the-art reasoning models like o1 and R1) constantly converges at a low accuracy that can be achieved with other simple scaling strategies (*e.g.* majority voting) under the same token budget. This suggests

that the model's underlying capability is not the primary bottleneck; rather, the way we orchestrate test-time compute can be improved.

Through a closer analysis of the reasoning process in LLMs, we find that the reasoning performance is often constrained by the model's initial thoughts, a phenomenon we refer to as *Tunnel Vision*. Specifically, the first few tokens generated in a Chain-of-Thought (CoT) can lock the model into a suboptimal reasoning path, preventing it from discovering more effective ideas in subsequent decoding steps.

Based on these insights, we argue that the reasoning process of LLMs should be executed in a parallel, multi-threaded manner. By ensuring each thinking thread operates independently, we can mitigate Tunnel Vision and foster a diversity of thought. Furthermore, parallel thinking offers significant deployment advantages, as the decoding process can be batched to better utilize memory bandwidth, which in turn leads to improved arithmetic intensity (Williams et al., 2009) (the ratio of floating-point operations to total data movement).

To put parallel thinking into practice, we introduce an end-to-end solution, ParaThinker, which enables native parallel thinking in LLMs by allowing the model to generate diverse thoughts and aggregate them into a final answer. The major challenges to develop ParaThinker include how to induce thought diversity and how to avoid thought conflict, which we address by introducing trainable control tokens to trigger distinct reasoning trajectories, thought-specific positional embeddings to distinguish different paths, and a two-phase attention mask design that enforces independence during reasoning and controlled integration during summarization. Specifically, our solution features three core innovations:

- **Specialized Control Tokens:** We introduce a set of trainable tokens (*e.g.* `<think i>`) to explicitly guide the model's generation. Each `<think i>` token prompts the model to initiate a distinct reasoning path, which ensures diversity in reasoning.
- **Thought-Specific Positional Embedding:** To resolve positional ambiguity when merging parallel thoughts, we augment the standard positional encoding with a unique, learnable embedding for each reasoning path. This allows the model to unambiguously differentiate the origin of each token during the final summarization stage.
- **SFT Training Pipeline:** We employ a scalable supervised fine-tuning (SFT) strategy where the model is trained on reasoning paths sampled from a teacher model. By randomly assigning the specialized `<think i>` tokens during this process, the model learns to generalize, enabling it to generate more parallel paths at inference time than were seen during training.

We evaluate ParaThinker on challenging math and coding benchmarks: AIME 2024, AIME 2025, AMC 2023, MATH-500 (Hendrycks et al., 2021), and LiveCodeBench v6 (Jain et al., 2024) against baselines such as standard autoregressive reasoning (DeepSeek-AI, 2025), majority voting (Chen et al., 2024a), and re-prefilling. Our approach demonstrates a remarkable leap in performance, achieving significantly improved accuracy with additional benefits of parallel computing. This efficiency allows smaller LLMs equipped with our native thought parallelism to outperform much larger, standard reasoning models, charting a new path for scaling test-time compute.

In summary, the contributions of our work are: (1) We characterize the test-time scaling bottleneck in LLM reasoning and attribute it to a narrow reasoning pathway, termed Tunnel Vision, which restricts the model's exploration during generation. (2) We propose and demonstrate that thought parallelism is a better way to scale LLM test-time compute. (3) We introduce an end-to-end solution to enable native parallel thinking. The resulting model, ParaThinker, achieves higher accuracy than sequential LLMs by 12.3% and 7.5% for 1.5B and 7B models, respectively. Compared with majority voting, ParaThinker further improves accuracy by 4.3% and 2.0%.

## 2 UNDERSTANDING THE SCALING BOTTLENECK

### 2.1 IS THE BOTTLENECK DUE TO LLM CAPABILITY OR SCALING STRATEGY?

To empirically ground our approach, we first characterize the limitations of conventional test-time scaling. We start by evaluating the DeepSeek-R1-distill-Qwen-1.5B model (DeepSeek-AI, 2025) on the AIME 2024 benchmark under various computational budgets. We control the budget by im-

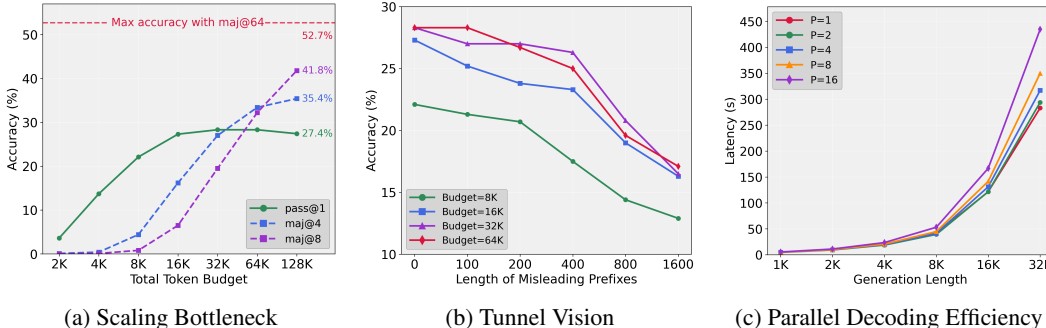

(a) Scaling Bottleneck       (b) Tunnel Vision       (c) Parallel Decoding Efficiency

Figure 1: Diagnosing the limitations of sequential reasoning and the potential of parallelism. All experiments use DeepSeek-R1-Distill-Qwen-1.5B on the AIME 2024 benchmark. (a) Scaling Bottleneck: Accuracy against the total number of token budget (for majority voting *e.g.* maj@4, the total token budget is the sum across all parallel paths). (b) Tunnel Vision: Ability to recover from its own potential mistakes with different lengths of misleading prefixes. The model generates solutions starting from flawed prefixes of length $n \in \{0, 100, \ldots, 1600\}$, denoting the first $n$ tokens of reasoning paths from the same model that previously resulted in a wrong answer. (c) Parallel Decoding Efficiency: Latency taken to decode $P \in \{1, 2, 4, 8, 16\}$ parallel paths, each of length $n \in \{1K, \ldots, 32K\}$.

posing a per-response token limit $B$ on the reasoning path. If the model fails to terminate naturally, we truncate the output and force termination by appending a terminal token (</think>). We also evaluate majority voting (Chen et al., 2024a; Fu et al., 2025) over $P \in \{4, 8, 16, 32, 64\}$ parallel samples, with each sample allocated $B/P$ tokens. For clarity, we plot only the results of $P = 4$ and $P = 8$ in the accuracy–budget curves, and report the maximum accuracy obtained with $P = 64$. The results, shown in Figure 1a, demonstrate that the performance of a single reasoning path (green line) quickly reaches the bottleneck, with additional tokens yielding negligible gains.

While some recent works attribute this phenomenon to "LLM overthinking" and attempt to solve it by compressing the model's output for more concise reasoning, these compressed models still encounter a bottleneck (Sun et al., 2025; Chen et al., 2025b; Fan et al., 2025; Cuadron et al., 2025). Different from these approaches, we investigate the fundamental cause of this bottleneck to determine how to further bootstrap LLM test-time scaling. And results in Figure 1a shows that majority voting can break through this bottleneck under the same total token budget, and the majority@64 with 2,048K total token budget (32K for each reasoning path), achieves a final accuracy far higher than the single-path approach. This significant gap suggests that *the bottleneck is not a hard limit of the model's reasoning capacity, but rather a symptom of the suboptimal test-time scaling strategy.* Simply allocating more test-time compute to a single-sequence LLM is not as effective as exploring multiple reasoning paths.

## 2.2 THE TUNNEL VISION OF SEQUENTIAL TEST-TIME SCALING

We hypothesize the bottleneck arises because an LLM's early token choices irreversibly commit it to a specific line of thought, making it difficult to escape initial errors. We call this Tunnel Vision: flawed initial reasoning locks the model into a suboptimal trajectory from which it cannot recover. To test this, we investigate the model's *recovery capacity* from erroneous starting points: For each AIME 2024 problem, we use DeepSeek-R1-Distill-Qwen-1.5B (DeepSeek-AI, 2025) to generate multiple samples. From the samples that produce incorrect answers, we extract prefixes of its flawed reasoning at lengths of 0, 100, 200, 400, 800, and 1600 tokens. We then prompt the model to continue generating from these erroneous prefixes and measure its final accuracy by sampling 16 times and calculating the average accuracy. The results, plotted in Figure 1b, show a clear negative correlation: the longer the erroneous prefix, the lower the final accuracy. This indicates that *the scaling bottleneck is a direct symptom of Tunnel Vision, where flawed initial tokens lock the model into a suboptimal reasoning path.* The longer the flawed prefix, the harder it is for the model to pivot to a correct solution, even with ample remaining budget.

## 3 WHY NATIVE THOUGHT PARALLELISM?

Given the limitations of sequential reasoning exposed by Tunnel Vision, we are motivated to explore parallel reasoning as a natural alternative. Diverse independent reasoning paths can potentially reduce the risk that the model becomes stuck in a single suboptimal trajectory. A commonly used practical instantiation of this idea is majority voting (Chen et al., 2024a), which aggregates many independent answers into a final decision.

However, majority voting is a heuristic—it applies the simple, fixed rule of choosing the most frequent answer without learning to evaluate the quality of the reasoning that produced it. This approach has two important drawbacks: **(1) Poor generalization:** Voting is only applicable to tasks with easily quantifiable answers (*e.g.*, multiple-choice or numeric values), failing on open-ended domains like complex agentic workflows, coding, or long-form proofs. **(2) Information loss:** Majority voting considers only the final answer, instead of the reasoning path, ignoring the valuable rationale, evidence, and intermediate steps within each reasoning path, making it impossible to synthesize insights from the full reasoning process to arrive at a better conclusion.

To overcome these limitations, we propose **native parallel reasoning** as an end-to-end solution. In this approach, we train the LLM to not only generate multiple reasoning paths in parallel but also to analyze on all the reasoning paths and generate the answer. Instead of relying on a fixed heuristic, the model learns to directly process and synthesize the full token trajectories of its parallel thoughts to produce a final, consolidated answer. This method makes the aggregation process itself trainable, allowing the model to learn optimal strategies for combining evidence, and path-aware, as it preserves the valuable information contained within each reasoning chain. We formalize *Why native parallel reasoning has the potential to outperform majority voting* theoretically in Appendix A.10.

Moreover, parallel decoding is highly efficient, as LLM inference is memory-bound (parameter and KV cache loading) rather than compute-bound (Sadhukhan et al., 2025). Batching $P$ paths amortizes memory accesses, enhancing arithmetic intensity and GPU utilization. Experiment with DeepSeek-R1-Distill-Qwen-1.5B using vLLM (Kwon et al., 2023) on an A800 GPU (Figure 1c) confirms: small $P$ incurs similar latency to a single path, while $P = 16$ is under $2\times$. This efficiency makes parallel exploration scalable for overcoming Tunnel Vision and boosting reasoning performance.

Prior works on parallel computation typically relies on external verifiers for search (Snell et al., 2024; Ghosal et al., 2025), which introduce a scalability bottleneck. Several concurrent works (Zhao et al., 2025b; Yang et al., 2025b; Zheng et al., 2025; Fu et al., 2025) have also investigated the parallel reasoning mechanism of LLMs, while they struggle to achieve substantial accuracy improvement due to positional ambiguity, computational inefficiency and/or limited thought diversity. More related works are discussed in Appendix A.2.

## 4 MODEL DESIGN

**Preliminaries of conventional sequential reasoning.** We denote an LLM by $\pi_\theta$, where $\theta$ is the set of model parameters. Given an input prompt of $l$ tokens $x = \{x_i\}_{i=1}^l$. The LLM then autoregressively generates an output sequence $y = (y_1, y_2, \ldots, y_L)$ with the conditional probability: $\pi_\theta(y|x) = \prod_{t=1}^L \pi_\theta(y_t|x, y_{<t})$. For tasks requiring multi-step reasoning, the output $y$ can be decomposed into a reasoning path $r$ followed by a final answer $a$: $y = (r, a)$. During decoding, each new token $y_t$ requires attention over the full context $x, y_{<t}$, which involves computing Key ($K$) and Value ($V$) tensors. To avoid recomputation when generating each $y_t$, LLMs often use a KV-cache to store $K$/$V$ tensors.

### 4.1 PARATHINKER WORKFLOW

As shown in Figure 2, our approach extends the sequential reasoning LLM paradigm by first generating a set of $P$ distinct reasoning paths $\{r^{(1)}, r^{(2)}, \ldots, r^{(P)}\}$ for a single input $x$ in parallel. Each individual reasoning path $r^{(i)}$ is a sequence of tokens representing a unique line of thought, sampled from the distribution: $\pi_\theta(r^{(i)}|x) = \prod_{t=1}^{L_i} \pi_\theta(r_t^{(i)}|x, s^{(i)}, r_{<t}^{(i)})$. Here, $s^{(i)}$ is a special control token that helps initiate a distinct reasoning path, which will be detailed in Section 4.2.

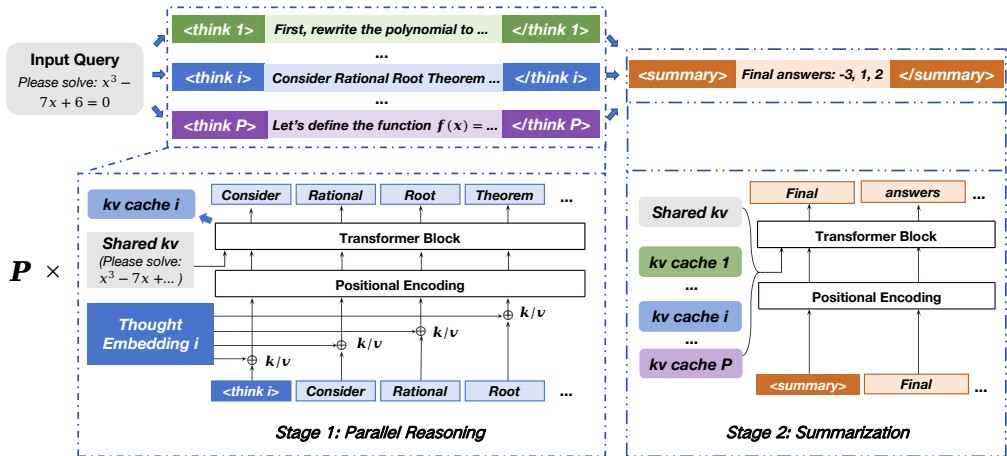

Figure 2: ParaThinker architecture. For an input question, ParaThinker processes it in two stages: (1) Parallel Reasoning: ParaThinker generates $P$ reasoning paths in parallel; (2) Summarization: ParaThinker merges the reasoning paths by reusing their KV-caches to generate the final answer.

After generating these parallel paths, the model synthesizes them to produce a final answer, $a$. This answer is conditioned on both the original prompt $x$ and the complete context of all preceding reasoning paths. Let $\mathcal{R} = (r^{(1)}, r^{(2)}, \ldots, r^{(P)})$ be the concatenation of all generated reasoning paths. The final answer $a$ is then sampled from the model as follows: $\pi_\theta(a|x) = \prod_{t=1}^{La} \pi_\theta(a_t|x, \mathcal{R}, a_{<t})$. Crucially, ParaThinker leverages the KV-caches from the parallel reasoning stage, eliminating the need to re-prefill the context and thereby offering significant computational savings compared to other methods.

## 4.2 SPECIAL TOKENS FOR BOOSTING THOUGHT DIVERSITY

ParaThinker needs to ensure diverse reasoning paths to avoid the trap of relying on a single sampled sequence. To achieve this, we introduce a set of trainable special control tokens: `<think i>`, `</think i>`, `<summary>`, and `</summary>` for $i \in \{1, \ldots, P\}$ to control the parallelization and merging operations. The `<think i>` token (denoted as $s^{(i)}$ in our equations) is placed at the beginning of each reasoning path, which leads the model to generate a distinct trajectory. Thus, the distribution of each reasoning path $\pi_\theta(r^{(i)}|x) = \prod_{t=1}^{L_i} \pi_\theta(r_t^{(i)}|x, s^{(i)}, r_{<t}^{(i)})$ will be conditioned by $s^{(i)}$. The closing `</think i>` token marks the end of a specific path, and the generation of the final answer is then wrapped within `<summary>` and `</summary>` tokens. This structured use of control tokens is a simple yet powerful mechanism to guide the model's generation process towards diverse and parallel lines of thought.

## 4.3 THOUGHT-SPECIFIC POSITIONAL EMBEDDING

Merging multiple reasoning paths poses challenges due to positional ambiguity. LLMs distinguish tokens based on their content and positional encoding. When multiple reasoning paths are generated in parallel, tokens at the same relative position (*e.g.* the $t$-th token in each reasoning path $r^{(i)}$) share identical positional encodings. This may cause confusion during summarization, as the model cannot differentiate which reasoning stream a token originated from.

**Flattened Encoding:** A naive solution assigns unique absolute positions across all paths: $m = l_x + i \cdot l_{\max} + t$, where $l_x$ is the input length, $i$ indexes the reasoning path, and $t$ indexes the token position within that path. While this resolves positional collisions, it results in large positional indices as $P$ increases. Typical positional encoding mechanisms such as Rotary Position Embedding (RoPE) (Su et al., 2024) encodes relative positions via rotations, and large index differences $|m - n|$ cause attention scores to decay. As a result, tokens from earlier paths (*i.e.* lower $i$ of $r^{(i)}$) contribute less when generating the final answer, introducing imbalance across paths.

**Sequence-Aware Positional Embedding:** To address positional ambiguity in multi-response generation tasks, ParaThinker separates different reasoning paths by augmenting the RoPE mechanism with learnable thought embeddings $\{T^{(j)}\}_{j=0}^P$. Specifically, we add the $T^{(j)}$ to the key and value embeddings of all tokens within the $i$-th reasoning path, which distinguishes each reasoning path at the summarizing phase. The thought embedding is added to the key before the RoPE rotation is applied. Let $\tilde{k}_t^{(j)}, \tilde{v}_t^{(j)}$ denote the cached key and value for token $t$ at path $j$, respectively, from which the key and value vectors are formed as:

$$\tilde{k}_t^{(j)} = R_t(k_t^{(j)} + T^{(j)}) \qquad \tilde{v}_t^{(j)} = v_t^{(j)} + T^{(j)} \tag{1}$$

Here, $l_{max}$ denotes the maximum token number for each reasoning path, and $R_t$ is the corresponding RoPE rotation matrix. Using the RoPE property $(R_n)^T R_m = R_{m-n}$, the dot product attention score between a query $q_n$ from the summary (at local position $n$) and a key $\tilde{k}_t^{(j)}$ from path $j$ (at position $m$) is:

$$\text{score}(n, m) = (R_n q_n^{(i)})^T \tilde{k}_m^{(j)} = (R_n q_n^{(i)})^T [R_m(k_m^{(j)} + T^{(j)})] = \underbrace{q_n^T R_{m-n} k_m^{(j)}}_{\text{Content-to-Content}} + \underbrace{q_n^T R_{m-n} T^{(j)}}_{\text{Content-to-Segment}}$$

$$\tag{2}$$

The Content-to-Content term is the standard RoPE attention score, which calculates the relevance between the query's content ($q_n$) and the key's content ($k_m^{(j)}$). This term is not related to the reasoning path number $j$ and thus does not change when scaling parallel reasoning paths. Content-to-Segment term calculates the relevance between the query's content ($q_n$) and the learnable identity of the key's entire reasoning path ($T^{(j)}$). This allows the query to directly probe for the origin of the information. Because each reasoning path has a unique, trainable thought embedding, this term provides an unambiguous signal for the model to differentiate between parallel streams of text, solving the positional ambiguity.

## 5 TRAINING AND DEPLOYMENT OF PARATHINKER

### 5.1 SCALABLE TRAINING DATA CURATION

ParaThinker models are trained by fine-tuning existing sequential reasoning LLMs with synthetic parallel thought data. We design a scalable training data curation pipeline that consists of two key components: multi-path training data scaling and extensible special tokens training.

**Multi-Path Training Data Scaling:** We develop a simple yet effective high-quality parallel reasoning dataset by sampling multiple times from teacher reasoning LLM (*e.g.* DeepSeek-R1 (DeepSeek-AI, 2025)). For a query $x$ and groundtruth answer $a$, let the $\hat{P}$ sampled answers denoted as $\{(r^{(1)}, a^{(1)}), (r^{(2)}, a^{(2)}), \ldots, (r^{(\hat{P})}, a^{(\hat{P})})\}$. We then concat each parallel answer into a groundtruth answer with the format of: $\hat{y} = ($`<think 1>`$r^{(1)}$`</think 1>`$, \ldots,$ `<think `$\hat{P}$`>`$r^{(\hat{P})}$`</think `$\hat{P}$`>` `<summary>`$a$`</summary>`$)$. The resulting $Data_{r=1}^{\hat{P}} = (x, \hat{y})$ pairs are then used for SFT.

**Extrapolative Special Tokens Training:** Due to the high cost of teacher LLM inference, we are often faced with the situation where we cannot generate enough reasoning paths when creating $Data_{sft}$, *i.e.*, $\hat{P} < P$, where $P$ denotes the maximum number of parallel reasoning paths supported at inference time. Thus, during SFT stage, LLMs have to learn to extrapolate to $(r^{(\hat{P}+1)}, \ldots, r^{(P)})$ with training data $Data_{r=1}^{\hat{P}}$. We develop a dynamic special token sampling method for extrapolative special tokens training. For each training batch, we randomly sample $\hat{P}$ special tokens from `<think i>`$, i \in \{1, \ldots, P\}$ and prepend these sampled tokens to the beginning of each reasoning sequence. Thus, each `<think_i>` specializes to induce diverse inference-time trajectories, despite training on only $\hat{P}$ paths.

### 5.2 TRAINING AND INFERENCE IMPLEMENTATION

**Attention Mask Design.** ParaThinker employs a two-phase attention mask design to enable parallel reasoning. In the reasoning phase, each path is decoded independently, with attention limited to

the input prompt and its own generated tokens, thereby preventing inter-path interference. In the summarization phase, answer tokens attend to the full prompt, all reasoning paths, and prior answer tokens, allowing integration across paths while preserving autoregressive consistency. More details are shown in Appendix A.3.

The inference engine for ParaThinker is built upon the vLLM framework (Kwon et al., 2023) to leverage its efficient PagedAttention mechanism for parallel decoding. The inference process is divided into two distinct phases:

**Parallel Reasoning Phase:** The engine processes the $P$ reasoning paths concurrently as a single batch. This parallel decoding phase terminates for all paths as soon as the first parallel reasoning path is completed, namely any of the $P$ paths generates an end-of-sequence (EOS) token. This termination strategy ensures all reasoning paths maintain an equal length, preventing processing imbalance. As empirically justified in Section A.8, this strategy yields the highest accuracy.

**Summarization Phase:** Following the parallel reasoning phase, the engine constructs an attention context spanning the KV caches of all $P$ reasoning paths, eliminating the need for costly re-prefilling. Leveraging vLLM's PagedAttention, this step is performed with zero data copying, as the summary sequence can directly reference the memory blocks of all preceding paths.

## 6 EXPERIMENTS

### 6.1 EXPERIMENTAL SETUP

**Training Details:** Our math reasoning experiments are based on a Qwen-2.5 (Qwen et al., 2025) 1.5B and 7B model distilled from DeepSeek-R1 (DeepSeek-AI, 2025) (which we denote as original R1-1.5B and R1-7B below), and coding experiments are based on DeepSeek-R1-Distill-Qwen-1.5B. For math reasoning, we construct a parallel reasoning dataset with 6.2K problem-solution pairs, with each instance consisting of a query ($x_i$), ground-truth answer ($a_i$), and $\hat{P} = 6$ distinct reasoning paths. During every training step, we randomly choose a path number $P$ from the set $\{2, 4, 6\}$ and construct a training sample by concatenating $P$ samples. For code reasoning, we construct a dataset of 50k problem–solution instances from OpenCodeReasoning (Ahmad et al., 2025), with questions that include more than 2 solutions. The final dataset includes either 2 or 4 alternative code-reasoning traces per problem ($\hat{P} \in \{2, 4\}$ during SFT). More details about training settings are listed in Appendix A.4.

**Baselines:** We compare ParaThinker against: *(1) Sequential:* Direct reasoning with original 1.5B/7B models. *(2) Majority Voting:* Generate $P$ independent paths and return the majority answer (Chen et al., 2024a). *(3) Re-Prefilling:* Generate $P$ paths, concatenate them, and feed the full context into the model for summarization. This mimics ParaThinker 's summarization but is inefficient since KV caches are not reused (Section 6.2). We do not include majority-voting for coding experiments since program correctness is often non-trivial to vote on.

**Benchmarks and Evaluation Setup:** We evaluate our model on 4 mathematical reasoning benchmarks: AIME 2024, AIME 2025, AMC 2023, and MATH-500 (Hendrycks et al., 2021), and 1 coding benchmark: LiveCodeBench v6 (Jain et al., 2024). We use a token budget control method where each reasoning path is limited to a maximum of $B$ tokens ($|r^{(i)}| \leq B$). If a model reaches this budget without naturally stopping, we enforce termination and then initiate the summarization stage by adding the (`<summary>`) token. This allows us to examine the utilization of the test-time scaling budget. More details of evaluation setup is shown in Appendix A.5.

### 6.2 SCALING PERFORMANCE

Table 1 and Table 2 compare ParaThinker with baseline methods under different token lengths. Compared with sequential LLMs, ParaThinker improves accuracy by up to 14.5% (1.5B) and 8.3% (7B) on AIME 2024, and by 3.6% (1.5B) and 8.8% (7B) on AIME 2025 at each token length on average (*e.g.* $2 \times 16K$ vs. 32K), demonstrating the effectiveness of parallel reasoning. This indicates that the summarization stage captures a richer aggregation strategy than vote counting. For coding tasks, ParaThinker outperforms original LLM and data ablation LLM (trained on the same dataset with ParaThinker) with $P$ increases, showing the potential to handle various tasks.

| Method | Base model: R1-1.5B | | | | Base model: R1-7B | | | |
|---|---|---|---|---|---|---|---|---|
| | AIME 24 | AIME 25 | AMC 23 | MATH | AIME 24 | AIME 25 | AMC 23 | MATH |
| Seq. (16K) | 26.1 | 22.4 | 67.1 | 81.2 | 51.9 | 37.9 | 88.4 | 91.2 |
| Seq. (32K) | 28.3 | 24.5 | 68.9 | 81.8 | 55.5 | 37.9 | 89.8 | 92.0 |
| Seq. (64K) | 27.1 | 25.5 | 67.7 | 81.7 | 56.0 | 39.6 | 89.8 | 92.5 |
| Seq. (128K) | 27.4 | 22.1 | 68.0 | 81.8 | 52.7 | 40.4 | 89.8 | 92.6 |
| Maj. (2×16K) | 25.9 | 23.0 | 67.0 | 81.4 | 52.3 | 38.3 | 88.4 | 91.4 |
| Maj. (4×16K) | 32.9 | **27.5** | 74.3 | 86.7 | 60.6 | 43.1 | 92.2 | 93.5 |
| Maj. (8×16K) | 41.0 | 31.8 | 79.8 | 89.0 | **68.8** | 49.6 | 93.1 | 94.2 |
| Rep. (2×16K) | 30.4 | 26.7 | 70.6 | 60.8 | 42.9 | 33.8 | 88.1 | 63.8 |
| Rep. (4×16K) | 24.2 | 25.8 | 61.3 | 58.6 | 43.3 | 33.3 | 86.3 | 63.2 |
| Rep. (8×16K) | 14.2 | 13.3 | 60.0 | 55.3 | 43.3 | 31.7 | 91.9 | 63.7 |
| **Para. (2×16K)** | **34.8** | **24.2** | **73.1** | **87.5** | **57.1** | **46.0** | **89.5** | **93.2** |
| **Para. (4×16K)** | **43.3** | **26.7** | **80.8** | **88.7** | **63.3** | **46.9** | **91.7** | **94.2** |
| **Para. (8×16K)** | **48.1** | **31.9** | **83.1** | **89.7** | **68.8** | **51.3** | **93.3** | **94.5** |

Table 1: Accuracy of ParaThinker and baselines (Seq., Maj., Rep. Para. denote Sequential, Majority Voting, Reprefill, and ParaThinker respectively). We report Pass@1 accuracy (%). Values in brackets indicate the maximum generation length $L$ (*e.g.* 16K); for parallel generation methods, we use $P \times L$ to denote generating $P$ reasoning paths, each with a maximum length of $L$.

| Model | Original Sequential LLM | | | | Fine-tuned Sequential LLM | | | | **ParaThinker-1.5B** | | |
|---|---|---|---|---|---|---|---|---|---|---|---|
| Budget | 12K | 24K | 48K | 96K | 12K | 24K | 48K | 96K | 2×12K | 4×12K | 8×12K |
| Accuracy | 18.3 | 18.9 | 18.9 | 17.7 | 16.5 | 16.0 | 16.0 | 16.7 | 18.7 | 19.4 | **20.1** |

Table 2: Pass@1 accuracy of ParaThinker, original sequential LLM, and the sequential LLM fine-tuned with the same data as ours.

.

We analyze how performance scales with the number of parallel paths in Figure 3 and 4, where increasing the path count consistently yields higher accuracy at larger generation budget. For sequential reasoning LLM ($P = 1$), expanding the token budget beyond 32K yields no further accuracy gains, whereas ParaThinker continues to improve. These results indicate that ParaThinker effectively extends the scaling law beyond the point where sequential reasoning models typically encounter a test-time scaling bottleneck.

We further analyze the relations between ParaThinker and majority voting in detail. The result is shown in Table 3. We find that ParaThinker does not conflict with the majority voting. Instead, it can be combined with majority voting to achieve higher accuracy. The highest accuracy of ParaThinker-1.5B+maj@8 can reach 66.7% and 60.0% on AIME 2024 with $P = 4$ and $P = 8$, gaining 23.4% and 11.9% accuracy improvements against pass@1. (See Table 9 for results on 7B+maj@k)

**Inference Efficiency.** Figure 5 shows the latency of ParaThinker under different total token budgets. Under a large budget (*e.g.* , 128K), the latency of high parallel size (*e.g.* , 8×16K) is much less than sequential scaling. This is because the decoding phase is typically bounded by memory bandwidth, and increasing the number of parallel reasoning paths does not increase data movement operations. Our experiments show that the reasoning latency on $128K$ can even be reduced by 66.8% with $8 * 16K$. The efficiency of our method gives us the proof that we can achieve greater accuracy through parallel scaling within acceptable inference latency.

## 6.3 ABLATION STUDY

**Train Data:** We test how much the performance gain of ParaThinker attribute to the training data. Specifically, we use the same data of ParaThinker (6 samples for each question) to finetune the original LLM, without changing any other setting. Table 5 shows that finetuning does not improve performance, with results even slightly worse than the original LLM. ParaThinker, on the other hand, outperforms the fine-tuned variant across all budgets, confirming its effectiveness.

|         | *P*=1 | *P*=2 | *P*=4 | *P*=8 |
|---------|------|------|------|------|
| *pass@1* | 26.1 | 34.8 | 43.3 | 48.1 |
| *maj@4*  | 32.9 | 42.5 | 53.0 | 56.3 |
| *maj@8*  | 41.0 | 50.1 | 61.7 | 59.9 |
| *maj@16* | **47.8** | **56.7** | **66.7** | **60.0** |

Table 3: Performance comparison of ParaThinker-1.5B with majority voting on AIME 2024. $P$: number of parallel reasoning paths; $maj@k$: majority voting with $k$ samples.

| P | Maj | SoN | M+S | Para. | pass@1 | pass@P |
|---|-----|-----|-----|-------|--------|--------|
| 2 | 26.7 | 33.3 | 33.3 | **34.8** | 26.3 | 35.8 |
| 4 | 30.0 | 38.4 | 39.2 | **43.3** | 24.6 | 49.2 |
| 8 | 35.8 | 40.8 | 40.0 | **48.1** | 22.8 | 56.7 |

Table 4: Ablation study for aggregation strategies on AIME 2024. $SoN$: Shortest-of-N; $M+S$: SoN if no majority identified; $Para.$: ParaThinker-1.5B

|                | A24 | A25 | AMC | MATH | Avg. |
|----------------|-----|-----|-----|------|------|
| *R1-1.5B-SFT (Same dataset with ParaThinker)* | | | | | |
| Seq. (16K)  | 26.3 | 18.5 | 66.0 | 81.1 | 48.0 |
| Seq. (32K)  | 22.9 | 22.1 | 64.1 | 77.6 | 46.7 |
| Seq. (64K)  | 25.8 | 17.3 | 62.2 | 77.6 | 45.7 |
| Seq. (128K) | 24.8 | 21.9 | 63.6 | 78.6 | 47.2 |
| Maj. (2×16K) | 26.0 | 18.1 | 66.3 | 81.0 | 47.9 |
| Maj. (4×16K) | 32.2 | 23.4 | 72.1 | 86.5 | 53.6 |
| Maj. (8×16K) | 42.5 | 27.1 | 79.8 | 89.2 | 59.7 |
| Rep. (2×16K) | 23.3 | 16.3 | 65.6 | 76.8 | 45.5 |
| Rep. (4×16K) | 15.0 | 11.7 | 55.6 | 70.6 | 38.2 |
| Rep. (8×16K) | 15.8 | 9.2 | 58.8 | 66.6 | 37.6 |
| *ParaThinker-1.5B* | | | | | |
| Para. (2×16K) | **34.8** | 24.2 | 73.1 | **87.5** | **54.9** |
| Para. (4×16K) | **43.3** | 26.7 | 80.8 | **88.7** | **59.9** |
| Para. (8×16K) | **48.1** | 31.9 | 83.1 | **89.7** | **63.2** |

Table 5: Train data ablation result: Pass@1 accuracy (%) of the 1.5B sequential LLM fine-tuned with the same dataset as ParaThinker.

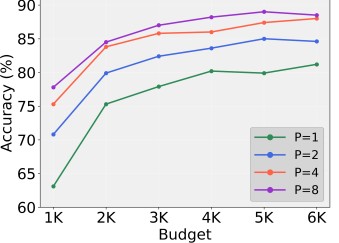

Figure 3: ParaThinker-1.5B Math-500 Scaling.

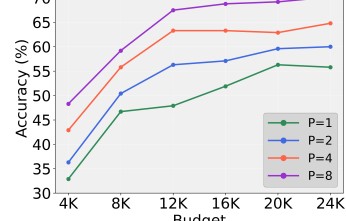

Figure 4: ParaThinker-7B AIME-24 Scaling.

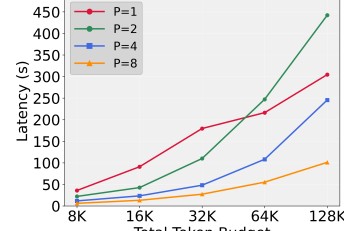

Figure 5: ParaThinker Latency under different budgets.

**Summarization Mechanism:** We compare our trainable summarization mechanism against alternative aggregation strategies like majority voting and a heuristic *Shortest-of-N*, which selects the final answer from the shortest reasoning path among the $N$ generated paths (motivated by the first termination strategy in ParaThinker). We also report the pass@P metric, which measures the fraction of problems solved correctly by at least one of the $P$ generated reasoning paths. We apply these strategies to the same set of reasoning paths generated by ParaThinker under a fixed budget of 16K tokens per path. As shown in Table 4, our end-to-end summarization outperforms all alternatives.

**Thought Embedding and Special Tokens:** We conduct ablation studies of thought embedding (by removing the thought embeddings) and special tokens (by replacing every `<think i>`, `</think i>` with `<think>`, `</think>`). We find that without special tokens and thought embeddings, the accuracy of ParaThinker drops 0.6%-3.1%, 1.4%-4.3% respectively, demonstrating the necessity of these designs, where detailed analysis are stated in Appendix A.9.

# 7 CONCLUSION

Our work identifies a limitation of conventional sequential test-time compute scaling of LLMs named "Tunnel Vision". We then introduce ParaThinker, a framework for native parallel reasoning that sidesteps the limitation by generating and aggregating multiple thought paths simultaneously. While our method presents a significant first step, future work could explore the interpretation of the learned aggregation strategy and end-to-end training with reinforcement learning.

## 8 REPRODUCIBILITY STATEMENT

For reproducibility, we provide detailed descriptions of our model architecture, training pipeline, and evaluation setup in Sections 4–6. Specifically, we describe the design of specialized control tokens, thought-specific positional embeddings, and the two-phase attention mask mechanism that enable native parallel reasoning. The training data curation process, including multi-path sampling from teacher models, extrapolative special token training, and dataset construction, is explained in Appendix A.4-A.6. Hyperparameters, optimization settings, and hardware configurations are also listed in Appendix A.4. All experiments are conducted with both 1.5B and 7B backbone models, and repeated across multiple reasoning budgets and path numbers to validate robustness. We further include ablation studies on positional embeddings, special tokens, termination strategies, and summarization methods to ensure the soundness of our conclusions. The implementation, datasets, and evaluation code will be released to the community to facilitate independent verification and further research.

## 9 ETHICS STATEMENT

Our study investigates methods to improve the reasoning efficiency and robustness of LLMs through parallel generation and aggregation of multiple reasoning paths. All benchmarks used in this work (AIME 2024, AIME 2025, AMC 2023, MATH-500, and LiveCodeBench v6) are publicly available, and no private or sensitive user data was involved. The proposed ParaThinker framework focuses on improving computational utilization and reasoning accuracy without altering semantic content in ways that could create additional risks of harmful or biased generation. While parallel reasoning may, in principle, be misused to amplify outputs, our design centers on controlled aggregation and transparent evaluation to mitigate such concerns. We adhere to the licensing requirements of all datasets and models used, and all data handling follows applicable privacy and legal standards. The authors declare no conflicts of interest or external sponsorship that could improperly influence this work. We encourage reviewer and community feedback on additional ethical considerations relevant to the deployment of parallel reasoning frameworks.

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

# A APPENDIX

## A.1 LLM USAGE STATEMENT

Large language models (*e.g.*, ChatGPT) were used only for minor language polishing and grammar correction. All ideas, experimental design, data analysis, and writing were conducted by the authors.

## A.2 RELATED WORKS

### A.2.1 SEQUENTIAL TEST-TIME SCALING

Recent advances in test-time scaling seek to improve LLM reasoning by increasing computational depth during decoding, primarily through reinforcement learning (RL) (OpenAI, 2024; DeepSeek-AI, 2025; Google, 2025; Yang et al., 2025a; Team et al., 2025; Zhao et al., 2025a; Yu et al., 2025) and supervised fine-tuning (SFT) (Muennighoff et al., 2025; Ye et al., 2025). RL-based methods (Ope-nAI, 2024; Shao et al., 2024) encourage LLMs to allocate more computation to promising reasoning paths by encouraging self-reflection and iterative trial. Other approaches distill long-form rationales from larger teacher models into smaller student models, enabling deeper internal reasoning through fine-tuned CoT supervision (Labs, 2025; Ye et al., 2025; Xu et al., 2025; Geiping et al., 2025). While these methods significantly enhance LLM performance on complex tasks, they often suffer from in-creased inference latency and compute consumption due to long output sequences (Sun et al., 2025; Zhu & Li, 2025; Qu et al., 2025; Wen et al., 2025). Moreover, excessively long reasoning traces may introduce "overthinking" effects such as repetition or hallucination (Chen et al., 2025b; Ghosal

et al., 2025). Besides, recent works have also shown that sequential reasoning LLMs are brittle to reasoning order (Chen et al., 2024b) or shallow token attacks (Xu et al., 2024; Qi et al., 2025). In contrast, our method introduces a new dimension of inference-time scaling—width—by executing multiple reasoning paths in parallel and summarizing them. This approach preserves reasoning efficiency while avoiding long single-path decoding.

### A.2.2 SEARCH-BASED METHODS FOR PARALLEL REASONING

Parallel Decoding LLMs improve reasoning by sampling multiple tokens at each step to accelerate LLM inference and/or improving LLM performance. Early techniques include beam search (Wiseman & Rush, 2016), self-consistency (Wang et al., 2023), speculative decoding (Leviathan et al., 2023; Chen et al., 2023) and majority voting (Chen et al., 2024a). Recent advancements include Best-of-N (Lightman et al., 2023), Tree of Thoughts (ToT) (Yao et al., 2023), and Monte Carlo Tree Search (MCTS) (Snell et al., 2024; Guan et al., 2025). These approaches typically require an external verifier to evaluate and rank candidate completions, increasing computational cost and often relying on domain-specific or manually constructed reward signals. Our method departs from these paradigms by generating multiple reasoning trajectories internally and merging them using a lightweight summarization step, without requiring external verifiers or retraining.

### A.2.3 NATIVELY PARALLEL GENERATION METHODS

Another line of work focuses on empowering LLMs to generate multiple tokens at each decoding iteration to accelerate LLMs theoretically. Diffusion-based language models (He et al., 2023; Ye et al., 2023; Zhao et al., 2025a; Wang et al., 2025; Arriola et al., 2025) sample multiple tokens in parallel during each diffusion step. While these methods can theoretically enable parallel generation, recent theoretical analyses (Feng et al., 2025) shows that for tasks involving sequential dependencies (*e.g.* reasoning), the number of required diffusion steps can scale linearly with sequence length, undermining their efficiency. PARSCALE (Chen et al., 2025a) investigates architectural parallelism by duplicating the input multiple times, applying distinct transformations, and aggregating outputs token-wise. However, this approach still requires architectural changes and specialized continual pretraining. In contrast, our approach retains the standard LLM architecture and introduces parallelism at the reasoning level by generating and caching multiple distinct chains of thought, which are later summarized into a final answer. Other works (Yang et al., 2025b; Pan et al., 2025; Rodionov et al., 2025; Jin et al., 2025) propose to automatically identify subtasks that can be solved in parallel. While effective for compositional tasks, it relies on explicit subtask decomposition, and these works focus on efficiency rather than accuracy improvement. ParaThinker, on the other hand, does not assume any subtask structure and improves both efficiency and accuracy by mitigating single-path failure cases (*e.g.* hallucinations or local optima) via diversity in reasoning. By integrating multiple KV caches in a summarization stage, our method scales inference without sacrificing correctness or requiring verifier models.

### A.3 ATTENTION MASK DESIGN

To enable efficient parallel reasoning in existing LLM infrastructures during both training and inference, ParaThinker adopts a two-phase attention mask design. During the reasoning phase, each reasoning path is decoded independently, with attention restricted to the input prompt and its own token history. Let $M_{i,j}$ denote the attention mask between the index $i$ and index $j$, where attention score can be calculated as: $A_{i,j} = \text{Softmax}\left(\frac{q_i \cdot k_j + M_{i,j}}{\sqrt{d_k}}\right)$. The attention mask for the $i$-th reasoning path ($r^{(i)}$) is defined as:

$$M_{t,j}^{r^{(i)}} = \begin{cases} 0, & \text{if } j \leq t \text{ and } j \in \{1, \ldots, l_x\} \cup \text{Ind}_i \\ -\infty, & \text{otherwise} \end{cases} \tag{3}$$

where $l_x$ is the length of the input prompt and $\text{Ind}_i$ is the index range for tokens in the $i$-th reasoning path. This enforces independence across reasoning paths by blocking inter-path attention.

During the summarization phase, where each answer token attends to the entire prompt, all reasoning paths, and previously generated answer tokens. The summarization attention mask is defined as:

$$M_{t,j}^A = \begin{cases} 0, & \text{if } j \leq t \text{ and } j \in \{1, \ldots, l_x\} \cup \bigcup_{i=1}^{P} \text{Ind}_i \cup \text{Ind}_a \\ -\infty, & \text{otherwise} \end{cases} \quad (4)$$

where $\text{Ind}_a$ denotes the index range of the answer tokens. This mask allows the final answer to integrate all parallel thoughts without violating autoregressive constraints.

## A.4 TRAINING DETAILS

This section details the configuration used for supervised fine-tuning (SFT) of the large language model.

**Dataset.** For math reasoning, 3.5K of the problems are sampled from the Open-R1 (Hugging Face, 2025) filtered to include only those with more than 4 existing answer variations. We also randomly sample 1.5K from and DeepMath (He et al., 2025) dataset, which provides 3 answers per question, and 1.2K from s1k (Muennighoff et al., 2025) (0.4K filtered for clear answers) and limo (Ye et al., 2025) (0.8K full dataset). To enrich diversity, we use gpt-oss-20b (OpenAI, 2025) as a teacher model, generating additional solutions at temperature 0.8, yielding six reasoning paths per problem.

| Parameter | Value |
|---|---|
| Batch Size | 1 |
| Gradient Accumulation Steps | 8 |
| Learning Rate | $1 \times 10^{-5}$ |
| Training Epochs | 3 |
| Context Length | 28,672 |
| Hardware | 4 GPUs |
| Learning Rate Scheduler | Constant |
| Warmup Ratio | 0.1 |
| Weight Decay | 0.05 |
| Max Gradient Norm | 0.5 |
| Max Context Length | 28K |

Table 6: SFT Training Configuration for DeepSeek-R1-Distill-Qwen-1.5B

| Parameter | Value |
|---|---|
| Batch Size | 1 |
| Gradient Accumulation Steps | 4 |
| Learning Rate | $2 \times 10^{-5}$ |
| Training Epochs | 2 |
| Context Length | 28,672 |
| Hardware | 8 GPUs |
| Learning Rate Scheduler | Cosine with Minimum LR |
| Warmup Ratio | 0.1 |
| Weight Decay | 0.05 |
| Max Gradient Norm | 0.5 |
| Max Context Length | 28K |

Table 7: SFT Training Configuration for DeepSeek-R1-Distill-Qwen-7B

## A.5 EVALUATION SETTINGS

Our system is implemented using the vLLM inference framework (Kwon et al., 2023), integrated with our custom parallel generation engine. We employ Qwen-2.5-math (Yang et al., 2024) and lighteval framework (Habib et al., 2023) for evaluation of math and code reasoning tasks respectively. For the 1.5B parameter model, we employ a sampling temperature of 0.5 and a top-p value of 1.0, while for the 7B parameter model, we use a temperature of 0.6 and a top-p value of 1.0. To account for output randomness, we report pass@1 accuracy, calculated as $pass@1 = \frac{1}{k}\sum_{i=0}^{k} p_i$, where $p_i$ is a binary indicator of correctness for the $i$-th response. Following DeepSeek-R1, we set $k$ depending on the size of test dataset, thus we set $k = 16$ for AIME 2024, AIME 2025, AMC 2023, and $k = 4$ for MATH-500. For coding tasks, we use $k = 8$ samples per test input when computing pass@1.

## A.6 EXAMPLES OF TRAINING DATA

After sampling multiple answers from the teacher model, we construct the SFT training data as follows: (1) concatenate all sampled answers as separate reasoning paths; (2) insert special tokens to delineate the parallel reasoning paths; (3) append the ground-truth final answer. An example is shown below:

---

**SFT Data Example**

<User>*Determine the number of solutions to the equation $z^{100} = 1 + i$ that lie in the third quadrant of the complex plane.* You FIRST think about the reasoning process as an internal monologue and then summarize the reasoning process to get the final answer. The summary process MUST BE enclosed within <summary> </summary> tags. The final answer MUST BE put in \boxed{}.<Assistant><think><vllm_pad>

<think1>
I need to solve $z^{100} = 1 + i$ and find how many solutions are in the third quadrant of the complex plane. First, I recall that complex numbers can be represented in polar form. So, I'll start by writing $1 + i$ in polar form ...

<think2>
Okay, so I need to find how many solutions of the equation $z^{100} = 1 + i$ are in the third quadrant. Hmm, let's start by recalling that complex equations like this one can often be solved using polar form and De Moivre's theorem. Let me jot down what I remember ...
</think2>

<think3>
Okay, so I need to figure out how many solutions of the equation $z^{100} = 1 + i$ lie in the third quadrant of the complex plane. Hmm, let's start by recalling that in the complex plane, the third quadrant is where both the real and imaginary parts of the complex number are negative. So any solution $z$ must satisfy $\text{Re}(z) < 0$ and $\text{Im}(z) < 0$ ...

<think4>
I need to solve the equation $z^{100} = 1 + i$ and find how many solutions are in the third quadrant of the complex plane. First, I recall that complex numbers can be expressed in polar form, which might help with exponents. So, let $z = re^{i\theta}$, where r is the modulus and $\theta$ is the argument. Then $z^{100} = r^{100}e^{i100\theta}$ ...
</think4>

<summary>
By analyzing multiple reasoning processes above, I concluded that: The final answer is $\boxed{25}$.
</summary>

---

## A.7 EXAMPLES OF PARALLEL ANSWERS

We illustrate the effectiveness of ParaThinker through an example. Question: "*Jen enters a lottery by picking* 4 *distinct numbers from* $S = \{1, 2, 3, \cdots, 9, 10\}$. 4 *numbers are randomly chosen from* $S$. *She wins a prize if at least two of her numbers were* 2 *of the randomly chosen numbers, and wins the grand prize if all four of her numbers were the randomly chosen numbers. The probability of her winning the grand prize given that she won a prize is* $\frac{m}{n}$ *where* $m$ *and* $n$ *are relatively prime positive integers. Find* $m + n$. " ParaThinker produces the following output (shortened for brevity):

> **Answer Example**
>
> <think1>
> Use the conditional probability formula
> $$P(A|B) = \frac{P(A \cap B)}{P(B)},$$
> where $A$ is winning the grand prize (all four numbers correct), and $B$ is winning a prize (at least two numbers correct). Calculate
> $$P(A) = \frac{1}{\binom{10}{4}} = \frac{1}{210}.$$
> Then calculate $P(B)$ as the probability of having at least two numbers correct, using the complement method:
> $$P(B) = 1 - P(0) - P(1) = 1 - \frac{\binom{6}{4}}{210} - \frac{\binom{4}{1}\binom{6}{3}}{210} = \frac{115}{210} = \frac{23}{42}.$$
> The final conditional probability is
> $$\frac{\frac{1}{210}}{\frac{23}{42}} = \frac{1}{115},$$
> hence $m + n = 116$.
> </think1>
>
> <think2>
> calculate $P(B)$ directly as the probability of having at least two numbers correct:
> $$\binom{4}{2}\binom{6}{2} + \binom{4}{3}\binom{6}{1} + \binom{4}{4}\binom{6}{0} = 115.$$
> The conditional probability is $\frac{1}{115}$
>
> <think3>
> Directly compute the intersection to confirm:
> $$P(A|B) = \frac{1}{115}.$$
>
> <think4>
> Use the hypergeometric distribution formula to calculate $P(B)$:
> $$P(B) = \sum_{k=2}^{4} \frac{\binom{4}{k}\binom{6}{4-k}}{\binom{10}{4}} = \frac{115}{210},$$
>
> <summary>
> By analyzing multiple reasoning processes above, I concluded that: The final answer is $\boxed{116}$.
> </summary>

This example demonstrates that the generated parallel answers employ diverse reasoning strategies that complement each other. By considering multiple reasoning paths, ParaThinker can mitigate the risk of being misled by a flawed initial line of thought.

|  | $P$=2 | $P$=4 | $P$=8 |
|---|---|---|---|
| Last-Finish | 32.1 | 37.1 | 42.5 |
| Half-Finish | 34.8 | 38.3 | 42.5 |
| First-Finish (Default) | 34.8 | 43.3 | 48.1 |

Table 8: Accuracy of ParaThinker-1.5B on AIME 2024 under budget $B$ for each reasoning path based on different strategies for terminating the parallel reasoning stage before proceeding to summarization.

|  | $P$=1 | $P$=2 | $P$=4 | $P$=8 |
|---|---|---|---|---|
| *pass@1* | 51.9 | 57.1 | 63.3 | 68.8 |
| *maj@4* | 60.6 | 64.9 | 70.3 | 74.7 |
| *maj@8* | 68.8 | 68.9 | 72.1 | 77.6 |
| *maj@16* | 73.3 | 70.0 | 73.3 | 76.7 |

Table 9: ParaThinker-7B together with majority voting on AIME 2024. $P$: number of parallel reasoning paths; $maj@k$: majority voting with $k$ samples.

## A.8 TERMINATION STRATEGIES FOR THE PARALLEL REASONING STAGE

We compare three strategies for terminating the parallel reasoning stage before proceeding to summarization: (1) *Last-Finish:* Wait for all $P$ paths to complete. (2) *Half-Finish:* Terminate when $P/2$ paths have completed. (3) *First-Finish:* Terminate when the first path completes (our default strategy).

As shown in Table 8, the First-Finish strategy yields the best performance. We attribute this to the fact that it maintains equal reasoning lengths across all paths, preventing any single path from dominating the context and ensuring a balanced contribution to the summarization stage. It is also, by definition, the most computationally efficient strategy.

## A.9 ABLATION

### A.9.1 TERMINATION STRATEGY ABLATION

We ablate the special control tokens (*e.g.* <think i>) and thought-specific positional embeddings in ParaThinker on the AIME 2024 benchmark using the 1.5B model with $P = 4$ parallel paths. Table 10 presents the results. We find that without special tokens and thought embeddings, the accuracy of ParaThinker drops 0.6%–3.1%, 1.4%–4.3% respectively, demonstrating the necessity of these designs for inducing thought diversity and resolving positional ambiguity during summarization

### A.9.2 FLATTENED POSITIONAL ENCODING ABLATION

We further ablate the use of flattened positional encoding (PE) as an alternative to our sequence-aware thought-specific PE, where unique absolute positions are assigned across all paths: $m = l_x + i \cdot l_{\max} + t$, with $l_x$ as the input length, $i$ indexing the reasoning path, and $t$ the token position within that path. Figure 6 shows the performance degradation as $P$ increases on AIME 2024. As illustrated, the Flatten-PE ParaThinker achieves high accuracy under low token budgets (e.g., approximately 70% at 1K tokens) but experiences a rapid decrease as the budget increases to 4K tokens, dropping to around 30%. In contrast, the original sequential approach (P=1) shows steady improvement over the same budget range. While this resolves positional collisions, it results in large positional indices as $P$ increases. Typical positional encoding mechanisms such as Rotary Position Embedding (RoPE) encodes relative positions via rotations, and large index differences $|m-n|$ cause attention scores to decay. As a result, tokens from earlier paths (*i.e.* lower $i$ of $r^{(i)}$) contribute less when generating the final answer, introducing imbalance across paths. Our thought-specific PE, by contrast, maintains balanced attention through learnable segment identities, yielding consistent gains.

|  | $P=2$ | $P=4$ | $P=8$ |
|---|---|---|---|
| **ParaThinker-1.5B** | 34.8 | 43.3 | 48.1 |
| Thought Embedding Ablation | 33.3 | 39.0 | 46.7 |
| Special Token Ablation | 34.2 | 40.2 | 45.2 |

Table 10: Ablation study on the effect of thought embedding (AIME 2024, avg@16, $t = 0.5$, $B = 16K$).

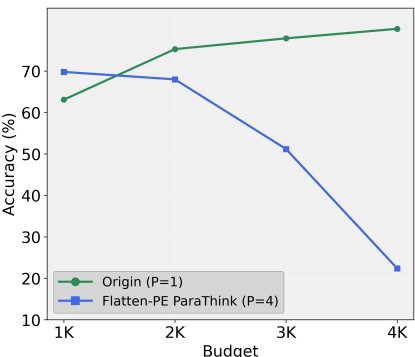

Figure 6: **Flattened PE Ablation:** Comparison on R1-1.5B trained with Flattened PE and original model (sequential) under Math-500 dataset.

### A.10 WHY PARATHINKER OUTPERFORMS MAJORITY VOTING

**Setup and notation.** Let $(q, R, a) \sim \mathcal{D}$, where $q$ is the input query, $R = (r^{(1)}, \ldots, r^{(P)})$ are $P$ reasoning paths (token trajectories), and $a$ is the true answer taking values in a finite set $\mathcal{A}$. Denote the data posterior by $p_{\text{data}}(a \mid q, R)$. Let $A = f(R) = (a^{(1)}, \ldots, a^{(P)})$ be the per-path answers extracted deterministically from $R$. An *aggregator* is any conditional distribution $p(a \mid q, R)$; we focus on two classes:

- **Voting-only aggregators:** $p(a \mid q, R)$ that depend on $R$ only through $A$, i.e. $p(a \mid q, R) = p(a \mid q, A)$.

- **Path-aware aggregators:** $p(a \mid q, R)$ that may use the full $R$.

**Markov chain view.** The dependencies among these variables can be expressed as the Markov chain

$$q \longrightarrow R \longrightarrow A \longrightarrow a.$$

Since $A$ is a deterministic function of $R$, and $a$ depends on the reasoning process through $R$, this chain emphasizes that conditioning on $A$ discards part of the information about $a$ contained in $R$. By the data-processing inequality,

$$I(a; R \mid q) \geq I(a; A \mid q),$$

with strict inequality whenever reasoning paths carry extra signal about $a$ beyond the final extracted answers.

**Information gap identity.** The conditional mutual informations satisfy the chain rule

$$I(a; R \mid q) = I(a; A \mid q) + I(a; R \mid A, q).$$

Since $A = f(R)$ is deterministic, $I(a; R \mid A, q) \geq 0$. Moreover, the conditional mutual information $I(a; R \mid A, q)$ admits the following exact representation as an expected KL:

$$I(a; R \mid A, q) = \mathbb{E}_{(q,A)} \mathbb{E}_{R|q,A} \Big[ D_{\text{KL}} \big( p_{\text{data}}(\cdot \mid q, R) \,\|\, p_{\text{data}}(\cdot \mid q, A) \big) \Big].$$

Equivalently (taken as expectation over $(q, R)$),

$$
I(a; R \mid q) - I(a; A \mid q) = \mathbb{E}_{(q,R)}\Big[ D_{\mathrm{KL}}\big( p_{\mathrm{data}}(\cdot \mid q, R) \,\|\, p_{\mathrm{data}}(\cdot \mid q, A) \big) \Big].
$$

We will call the left-hand side the *information gap*.

**Optimal KL for voting-only aggregators.** For any voting-only aggregator $p(a \mid q, A)$, the expected forward KL to the data posterior is

$$
\mathbb{E}_{(q,R)}\Big[ D_{\mathrm{KL}}\big( p_{\mathrm{data}}(\cdot \mid q, R) \,\|\, p(\cdot \mid q, A) \big) \Big].
$$

For each $(q, A)$, the choice of $p(\cdot \mid q, A)$ minimizing the inner expectation over $R \mid q, A$ is precisely $p_{\mathrm{data}}(\cdot \mid q, A)$. Therefore the minimal achievable expected KL over all voting-only aggregators equals the information gap:

$$
\inf_{p(\cdot \mid q, A)} \mathbb{E}_{(q,R)}\Big[ D_{\mathrm{KL}}\big( p_{\mathrm{data}}(\cdot \mid q, R) \,\|\, p(\cdot \mid q, A) \big) \Big]
$$

$$
= \mathbb{E}_{(q,R)}\Big[ D_{\mathrm{KL}}\big( p_{\mathrm{data}}(\cdot \mid q, R) \,\|\, p_{\mathrm{data}}(\cdot \mid q, A) \big) \Big]
$$

$$
= I(a; R \mid q) - I(a; A \mid q).
$$

Thus the information gap is an exact lower bound on the expected KL that any voting-only aggregator must incur. In fact, this bound is only attained by the Bayes-optimal choice $p(a \mid q, A) = p_{\mathrm{data}}(a \mid q, A)$, which requires full knowledge of the data distribution. Heuristic rules such as majority voting are far more restrictive and generally achieve strictly larger expected KL.

**ParaThinker (KL minimizer) vs. voting.** Let $\mathcal{F} = \{ p_\theta(a \mid q, R) : \theta \in \Theta \}$ be a family of path-aware aggregators (e.g., parameterized models trained with SFT on full paths). Define

$$
\theta^\star = \arg\min_{\theta \in \Theta} \mathbb{E}_{(q,R)}\Big[ D_{\mathrm{KL}}\big( p_{\mathrm{data}}(\cdot \mid q, R) \,\|\, p_\theta(\cdot \mid q, R) \big) \Big].
$$

Two observations follow immediately:

1. If $\mathcal{F}$ contains the Bayes-optimal voting aggregator $p_{\mathrm{data}}(\cdot \mid q, A)$ as a special case (i.e., some $p_\theta(\cdot \mid q, R) = p_{\mathrm{data}}(\cdot \mid q, A)$ for all $(q, R)$), then by minimality of $\theta^\star$,

$$
\mathbb{E}\big[ D_{\mathrm{KL}}(p_{\mathrm{data}} \| p_{\theta^\star}) \big] \leq \inf_{p(\cdot \mid q, A)} \mathbb{E}\big[ D_{\mathrm{KL}}(p_{\mathrm{data}} \| p(\cdot \mid q, A)) \big] = I(a; R \mid q) - I(a; A \mid q).
$$

2. If $\mathcal{F}$ is sufficiently expressive to approximate $p_{\mathrm{data}}(\cdot \mid q, R)$ well, then the left-hand side above can be made small (approaching zero), whereas the right-hand side equals the information gap and is strictly positive whenever $I(a; R \mid A, q) > 0$.

Consequently, in any non-degenerate situation where intermediate reasoning tokens in $R$ carry information about $a$ beyond $A$, path-aware SFT (ParaThinker) can attain strictly lower expected KL than any voting-only aggregator.

**From KL to 0–1 risk.** Define the expected classification error (0–1 risk) of an aggregator $p$ by

$$
\mathcal{R}(p) := \Pr_{(q,R,a) \sim \mathcal{D}} \big[ \hat{a} \sim p(\cdot \mid q, R) \text{ s.t. } \hat{a} \neq a \big].
$$

Pinsker's inequality implies, for each $(q, R)$,

$$
\mathrm{TV}\big( p_{\mathrm{data}}(\cdot \mid q, R), \, p_{\theta^\star}(\cdot \mid q, R) \big) \leq \sqrt{ \tfrac{1}{2} D_{\mathrm{KL}}\big( p_{\mathrm{data}}(\cdot \mid q, R) \,\|\, p_{\theta^\star}(\cdot \mid q, R) \big) }.
$$

Using the fact that the per-context increase in 0–1 risk is bounded by this TV distance, and taking expectation, we obtain

$$
\mathcal{R}(p_{\theta^\star}) \leq \mathcal{R}(p_{\mathrm{data}}) + \sqrt{ \tfrac{1}{2} \mathbb{E}_{(q,R)}\Big[ D_{\mathrm{KL}}\big( p_{\mathrm{data}}(\cdot \mid q, R) \,\|\, p_{\theta^\star}(\cdot \mid q, R) \big) \Big] }.
$$

Combining this with the KL comparison above gives the desired qualitative statement: when the information gap is positive, a path-aware aggregator that closely fits the data posterior will achieve lower 0–1 risk than any voting-only aggregator.

**Remarks and caveats.**

- The main identity linking mutual information and expected KL is exact and does not rely on asymptotics; it follows from the conditional mutual information representation $I(X; Y \mid Z) = \mathbb{E}_Z \mathbb{E}_{Y|Z} D_{\mathrm{KL}}(P_{X|Y,Z} \| P_{X|Z})$.

- Assumptions: we assumed a finite label alphabet for clarity (so KL and TV are finite); the same arguments extend to standard measurable settings with appropriate integrability conditions.

- Practical caveats: the inequalities above compare *best-possible* elements of model families. In practice, finite training data, limited model capacity, and optimization error mean $\theta^\star$ may not reach the theoretical minimum. Nonetheless, the direction of the inequality indicates when and why Path-aware SFT (ParaThinker) should outperform majority voting.

- Pinsker's inequality is loose; for numerical guarantees one may replace it with bounds tailored to the label loss or use calibrated surrogate-loss analyses.

**Connection to ParaThinker design.**    ParaThinker explicitly trains an aggregator to condition on full paths $R$, thereby directly targeting the KL objective minimized by $\theta^\star$ above. The identity and minimax reasoning explain why conditioning on full reasoning paths recovers information that voting discards.

