# OpenReview forum: "ParaThinker: Native Parallel Thinking as a New Paradigm to Scale LLM Test-time Compute"
_ICLR.cc/2026/Conference — Submitted to ICLR 2026_

### Official Review · Reviewer_iDsw · 2025-10-17

**Soundness:** 2
**Presentation:** 3
**Contribution:** 2
**Rating:** 4
**Confidence:** 4

**Summary:**

This paper first analyzes the shortcoming of current paradigm of sequential CoT scaling, that is, the bottleneck to the model's reasoning performance during CoT scaling is not the model's inherent reasoning capability, but the model’s imperfect initial steps predicted at the beginning. Then, the authors propose ParaThinker, a pipeline allowing LLMs to first generate multiple reasoning paths in parallel, and aggregate them to produce a final answer. By doing so, it makes the model fully explore the reasoning space and think more effectively. The experiments on some typical math reasoning benchmarks and the ablation studies help to validate the effectiveness of the proposed method.

**Strengths:**

(1) The studied problem, test-time scaling performance of LLMs, is a very hot and important topic.

(2) The paper is generally well-written, the motivation is clear and logically sound.

(3) The method that first performs parallel thinking then aggregates the answer is very straight-forward.

(4) I appreciate the design of ablations, I can see that the authors make great efforts on showing the effectiveness of each part in the method.

**Weaknesses:**

(1) The overall contribution is somehow limited. The proposed method is equivalent as first sampling multiple solutions in parallel, then using the model itself to summarize the most reliable answer from all generated solutions (where the summarization capability is specifically optimized). I can see that the authors make some improvements on the sampling efficiency issue, but the pipeline is not novel to me. Correct me if there are misunderstandings.

(2) The number of evaluation datasets and models is limited. For datasets, the sample sizes of AMC23 and AIME24-25 are quite small. The authors should include more datasets such as MinervaMath or OlympiadBench. Also, the authors only perform experiments on two R1-Distilled-Qwen models. I expect to see more results on other model variants, especially non-thinking models.

(3) The current method only explores the effectiveness of SFT paradigm, which relies on a teacher model for generating expert traces. How effective is it to further use reinforcement learning? Conduct RL experiments would further strengthen the paper.

**Questions:**

(1) When generating solutions with DeepSeek-R1 during SFT data generation, if all the sampled solutions for a given problem from DS-R1 are incorrect, will they still be concatenated with the ground-truth answer to form SFT samples? Do you perform extra data filtering and processing practice? Also, from the example in Appendix A.6, we can see that the final summarization does not include CoT process, while only concludes with the final answer. This makes me confused that   where does ParaThinker’s huge advantage over first sampling then majority voting come from? Could the authors provide some insights on this?

(2) The captions of table should be placed above the tables, according the submission guidelines.

---

### Official Review · Reviewer_Xkgn · 2025-10-31

**Soundness:** 1
**Presentation:** 3
**Contribution:** 1
**Rating:** 2
**Confidence:** 5

**Summary:**

This paper explores scaling compute in language models via parallel reasoning, training LLMs to natively generate diverse thought threads distinguished by trainable < think i > tokens, to mitigate "Tunnel Vision" (early flawed steps locking suboptimal paths). ParaThinker enables P independent paths via special tokens for diversity, thought-specific positional embeddings to resolve merging ambiguity, and a two-phase attention mask for independent generation followed by trainable synthesis. Trained via SFT on teacher-sampled multi-path data with dynamic token assignment for extrapolation, it yields accuracy gains on math/coding benchmarks under fixed token budgets, outperforming sequential baselines and majority voting and latency reductions via batched decoding.

**Strengths:**

- Parallelism scaling, as shown in previous works, is a powerful scaling dimension for language models, and this paper explores an area which is a promising direction for scaling the compute of language models.
- Comprehensive appendix giving examples of answer generation and methodology additional details.

**Weaknesses:**

- Notable absence of references to previous parallel work, particularly the omission of key prior art such as "Instilling Parallel Reasoning Into Language Models" (Macfarlane et al., 2025), which directly investigates native parallel reasoning in LLMs with mechanisms for diversity via short approach summaries and parallelization of decomposable operators. The paper also oversells the contributions with continued use of the wording "new paradigm" when this paradigm is well established and there are many works exploring it in these exact domains.

- The idea that you can enforce diversity with different numbered tokens is something I am very skeptical of. This paper claims that all distributions will lead to certain parts of the distribution being assigned to 1, 2, 3, 4, 5, etc., such that diversity can be achieved simply by sampling from the each token number. This mapping does not seem tractable to me. Previous papers such as Macfarlane et al. (2025) however have already tackled this by generating short instructions for each thread, an approaches that then guarantees threads to be diverse.

- The paper does not give results for Para 1x16k to check whether fine-tuning has boosted performance which should be included in the main results table

- Performance is tested on limited datasets.

- It only tests on a single model family, which is not enough to validate this as a general method.

- No reproduction of previous parallelism decoding methods; only simple baseline of Reprefill and majority voting, which are not competitive and do not reflect current top-performing methods.

- Lack of ablation checking that the tokens meaningfully increase diversity.

[1] Macfarlane et al., "Instilling Parallel Reasoning into Language Models," 2nd AI for Math Workshop @ 42nd International Conference on Machine Learning.

**Questions:**

- The use of numbered control tokens < think i > to enforce thought diversity is intriguing but raises concerns about whether this reliably samples distinct parts of the model's distribution. Could you please perform an experiment to validate that these tokens lead to meaningfully different reasoning paths? A small number of decoding examples is not sufficient to assess this.

- Could you discuss how in principle it would be possible for a neural network to learn for all distributions a separation of high likelihood trajectories to be tired to certain token numbers and how learnable this operation would be in practice.

- How does the performance compare with existing native parallelism approaches ,including Macfarlane et al which generates thread instructions as opposed to thread ids, to accept this paper such baselines and previous approaches to parallelism need to be added as baselines.

---

### Official Review · Reviewer_6w1C · 2025-11-03

**Soundness:** 3
**Presentation:** 3
**Contribution:** 3
**Rating:** 4
**Confidence:** 3

**Summary:**

The paper proposes ParaThinker, a framework that scales LLM reasoning not by generating longer sequential chains of thought, but by producing multiple diverse reasoning paths in parallel and then synthesizing them into a final answer. The method introduces special  tokens, thought-specific positional embeddings, and a two-phase attention mask to ensure reasoning paths are independent during generation and combined effectively during summarization.

**Strengths:**

* The paper moves beyond naïve parallel sampling and proposes an end-to-end framework where the model actually learns to generate multiple reasoning paths in parallel and then summarize them into a single final answer. This makes the approach more principled and model-driven, rather than relying on simple heuristics like majority voting.
* The authors also pay attention to real-world deployment. By reusing KV-caches and using batch decoding, the method reduces latency during inference, making parallel reasoning not only more accurate but also practical to use in real systems.

**Weaknesses:**

* There is not much analysis on why the parallel paths actually produce diverse and different reasoning paths. It’s not very clear whether the model really learns diversity, or if it’s just doing something similar to normal multi-sampling.
Because approach utilized large model multiple sampling where dataset doesn’t have parallel reasoning properties.
* The paper claims that diversity comes from using special tokens, but it doesn’t show whether the reasoning paths are truly different, or if they sometimes collapse into similar traces. It might need more evidence or some kind of regularization discussion.
* The method also seems quite dependent on teacher-generated data. Since all the multi-path samples come from the teacher model, it’s hard to tell how much the model is learning parallel thinking. It would be great if author compare the SFT model with teacher’s data and parallel thinking model on all the tasks.
* Currently, most of comparision is between base model and further trained model with distilled data, it is difficult to understand the gain is coming from parallel thinking path or distillation.

**Questions:**

What are the main differences with following work?

[1] Instilling Parallel Reasoning into Language Models

[2] Learning Adaptive Parallel Reasoning with Language Models

---

### Official Review · Reviewer_gKeT · 2025-11-03

**Soundness:** 3
**Presentation:** 2
**Contribution:** 2
**Rating:** 2
**Confidence:** 4

**Summary:**

This paper introduces ParaThinker, a novel test-time reasoning framework designed to overcome the "Tunnel Vision" bottleneck in large language model reasoning. Instead of relying on a single sequential chain-of-thought, ParaThinker explicitly generates multiple diverse reasoning paths in parallel using trainable <think i> control tokens and then summarizes them into a final answer using a dedicated decoding stage. Key innovations include specialized control tokens, thought-specific positional embeddings, and a two-stage attention mechanism that ensures path independence during reasoning and integration during summarization. Experiments on math (AIME, AMC, MATH-500) tasks demonstrate that ParaThinker consistently improves accuracy under fixed compute budgets, outperforming sequential, majority-voting, and reprefill baselines.

**Strengths:**

1. The paper is well-motivated and clearly written, and flows well from problem identification to method to experiments. The Tunnel Vision phenomenon makes sense, providing an intuitive angle to understand test-time scaling bottlenecks in LLMs.

2. The method components - trainable <think i> tokens, thought-specific embeddings, and a dual-phase attention mask—are well-integrated, requiring minimal changes to standard Transformers. The design choices are sensible and practically motivated.

3. On math benchmarks including AIME, AMC and MATH500, ParaThinker significantly outperforms sequential CoT and majority voting under fixed decoding budgets. The efficiency gains from parallel decoding are especially compelling.

**Weaknesses:**

1. Although LiveCodeBench is explicitly mentioned in the introduction and experiment setup, the paper does not provide any quantitative results for it. This is a critical omission, especially since the method claims to generalize to coding tasks. Were these experiments conducted? If yes, please report them. If not, the paper should not imply that coding performance has been evaluated.

2. Table 1 is labeled as reporting results on the “base model” (R1-1.5B), while Table 5 is meant to show ablation results on the same model fine-tuned with the ParaThinker dataset (SFT). However, the reported numbers in both tables are exactly the same. Could authors clarify this?

3. Recent works have also explored parallel reasoning with similar token-based methods. Although some of these are cited in the paper, they are not included as empirical baselines. Without such comparisons, it is hard to determine whether ParaThinker meaningfully advances the state of the art in this space.

4. The section title “Scalable Training Data Curation” focuses on synthetic data generation and sampling strategies rather than true dataset curation. A more accurate title would better reflect the scope of the section.

**Questions:**

Please refer to the Weaknesses section above.

---

### Author Response · Authors · 2025-12-04
**General Response to All Reviewers**

Thank you to the reviewers and area chair for the time and effort invested in handling this submission.

## Summary of Reviewers’ Noted Strengths
- The paper provides a clear diagnosis of the “Tunnel Vision” failure of sequential test-time scaling and introduces a concrete alternative via *native thought parallelism*.
- ParaThinker presents a coherent end-to-end design—special control tokens, thought-specific positional embeddings, and a two-phase attention mask—as a unified mechanism.
- Strong empirical gains are demonstrated across challenging math and coding benchmarks, with notable efficiency benefits due to KV-cache reuse.
- The paper includes extensive ablations and a formal information-theoretic justification showing why conditioning on full reasoning paths can outperform majority voting.

## Responses to Reviewers’ Concerns

### 1. LiveCodeBench Results
LiveCodeBench **was** included in the paper (Table 2). The omission was purely editorial: the caption did not explicitly restate the benchmark list. We will correct the caption to avoid any misunderstanding.

### 2. Table 1 vs. Table 5 “Identical Results”
The tables evaluate **different settings**:
- Table 1: base model across reasoning budgets.
- Table 5: *R1-1.5B-SFT* ablation models trained on the same multi-path dataset.

**The numbers are not the same**; the confusion likely stems from misreading.

### 3. Diversity From Special Tokens vs. Multi-Sampling
While explicit diversity metrics were limited, the paper already shows special token ablation in Table 10:
- Special-token removal **reduces performance** across multiple settings, indicating a causal contribution to effective parallel paths.

We will add explicit diversity measurements (path entropy, trace distance) in the revision for added clarity.

### 4. Gains From Architecture vs. Distilled Teacher Data
Table 5 already provides the requested control: the sequential model **fine-tuned on the exact same multi-path dataset** (R1-1.5B-SFT) with baseline method does **not** recover ParaThinker’s gains.
This demonstrates that improvements arise from the **architecture**, not merely from distilled data.

### 5. Differences From Prior Works
**Instilling Parallel Reasoning Into Language Models (Macfarlane et al.)**
- Not an end-to-end model; relies on stitching/re-prefill and does not reuse per-path KV caches.
- ParaThinker performs **native**, fully end-to-end parallel generation and summary with KV reuse, enabling substantially better efficiency.

**Learning Adaptive Parallel Reasoning (APR)**
- APR decomposes problems via explicit spawn/join logic or RL-based orchestration; effectiveness depends on problem decomposability.
- ParaThinker is task-agnostic, scalable to *arbitrary parallel size*, and does not rely on decomposability or RL.
We will expand related work to explicitly highlight these differences.

### 6. “Equivalent to Sampling + Summarization” (Novelty Concern)
ParaThinker introduces three tightly coupled innovations beyond naïve sampling:
1. **Native parallel generation** enforced by a two-phase attention mask. Reuse KV cache directly without reprefill for better efficiency.
2. **Thought-specific positional embeddings** preventing positional interference across paths for native parallel thinking.
3. **End-to-end trainable summarizer** conditioning on full reasoning trajectories.
Appendix A.10 further proves why path-aware aggregation strictly dominates majority voting.
This goes well beyond simple “sample then summarize.”

### 7. Data Quality, Incorrect Teacher Traces, and Majority Voting
Appendix A.4–A.6 describe filtering procedures and how paths are concatenated with ground truth. Even if some paths are incorrect, the summarizer is trained to integrate information from *all* trajectories. Appendix A.10 explains, from an information-theoretic perspective, why ParaThinker can exceed the performance ceiling of majority voting.

## Closing Remark
Most concerns (coding evaluation, data vs. architecture, special-token importance) are **already addressed** through existing tables, ablations, and theory. We will revise the paper to:
- Clarify captions (especially Table 2, Table 1 vs. Table 5),
- Add diversity analyses
- Strengthen the related-work comparisons.

We respectfully request reviewers to reassess their evaluations in light of these clarifications and the evidence already present in the submission.

---

### Meta-Review · Area_Chair_J2wD · 2026-01-04

**Summary:**

This paper proposes ParaThinker, a framework that replaces sequential test-time reasoning with native parallel generation of multiple reasoning paths followed by a learned aggregation step. Reviewers generally found the motivation intuitive and the system design coherent, and several acknowledged strong empirical gains on math benchmarks and reasonable efficiency considerations. However, across reviews there was substantial skepticism about the actual novelty of the approach, with multiple reviewers characterizing it as close to parallel sampling plus summarization, and questioning the repeated framing as a “new paradigm.”

The most influential concerns were the lack of direct and competitive baselines against prior parallel reasoning work, especially Macfarlane et al., limited evidence that the proposed control tokens truly induce meaningful diversity, and insufficient analysis disentangling architectural gains from those due to distilled teacher data. Additional issues such as unclear or missing LiveCodeBench reporting, confusing tables, and limited model/dataset coverage further weakened confidence in the results.

Based on these concerns and the overall reviewer sentiment, I recommend rejection. While the direction is promising, the current version does not yet make a sufficiently clear or well-supported contribution relative to existing work. I encourage the authors to significantly strengthen comparisons, tone down novelty claims, and add concrete diversity and ablation analyses before resubmission.

**Reviewer Concerns:**

There was no rebuttal provided for this submission. As a result, none of the substantive reviewer concerns were directly addressed during the discussion phase. This notably includes questions about missing or unclear coding results, table inconsistencies, and whether certain experimental claims were editorial errors or deeper issues.

Core concerns therefore remain outstanding. These include doubts about novelty beyond sampling-and-aggregation, lack of strong baselines to prior native parallel reasoning methods, insufficient evidence that the special tokens induce stable and diverse reasoning paths, and limited evaluation scope across models and datasets. Concerns about overselling the contribution and terminology (“new paradigm”) also remain unresolved.

**Reviewer Scores:**

Reviewer gKeT scored the paper as 2 (reject) and raised concerns about missing results, table inconsistencies, and limited comparisons; without a rebuttal, this score would almost certainly remain unchanged. Reviewer Xkgn also gave a 2 with strong confidence, citing lack of novelty and missing baselines, and there is no indication their assessment would soften without new evidence.

Reviewer 6w1C and reviewer iDsw both gave 4 (borderline reject), explicitly stating they would not mind acceptance but highlighting concerns around diversity, distillation dependence, and scope. In the absence of a rebuttal addressing these points, I expect their scores would stay at 4 rather than move upward. Overall, the score distribution would likely remain the same after discussion, reinforcing a reject decision.

---

### Decision · Program_Chairs · 2026-01-26

Reject